# Transferability of Cardiopulmonary Parameters between Treadmill and Cycle Ergometer Testing in Male Triathletes—Prediction Formulae

**DOI:** 10.3390/ijerph19031830

**Published:** 2022-02-06

**Authors:** Szczepan Wiecha, Szymon Price, Igor Cieśliński, Przemysław Seweryn Kasiak, Łukasz Tota, Tadeusz Ambroży, Daniel Śliż

**Affiliations:** 1Department of Physical Education and Health, Faculty in Biala Podlaska, Jozef Pilsudski University of Physical Education in Warsaw, 21-500 Biala Podlaska, Poland; szczepan.wiecha@awf.edu.pl (S.W.); igor.cieslinski@awf.edu.pl (I.C.); 23rd Department of Internal Medicine and Cardiology, Medical University of Warsaw, 02-091 Warsaw, Poland; szymonprice@gmail.com; 3Students’ Scientific Group of Lifestyle Medicine, 3rd Department of Internal Medicine and Cardiology, Medical University of Warsaw, 02-091 Warsaw, Poland; przemyslaw.kasiak1@gmail.com; 4Department of Physiology and Biochemistry, Faculty of Physical Education and Sport, University of Physical Education in Krakow, 31-571 Krakow, Poland; lukasz.tota@awf.krakow.pl; 5Institute of Sports Sciences, University of Physical Education in Krakow, 31-541 Kraków, Poland; tadek@ambrozy.pl; 6Public Health School Centrum Medyczne Kształcenia Podyplomowego (CMKP), 01-826 Warsaw, Poland

**Keywords:** exercise testing, sports diagnostic, maximal oxygen uptake, HR_max_, VO_2max_, triathletes, cardiorespiratory fitness, prediction models

## Abstract

Cardiopulmonary exercise testing (CPET) on a treadmill (TE) or cycle ergometry (CE) is a common method in sports diagnostics to assess athletes’ aerobic fitness and prescribe training. In a triathlon, the gold standard is performing both CE and TE CPET. The purpose of this research was to create models using CPET results from one modality to predict results for the other modality. A total of 152 male triathletes (age = 38.20 ± 9.53 year; BMI = 23.97 ± 2.10 kg·m^−2^) underwent CPET on TE and CE, preceded by body composition (BC) analysis. Speed, power, heart rate (HR), oxygen uptake (VO_2_), respiratory exchange ratio (RER), ventilation (VE), respiratory frequency (fR), blood lactate concentration (LA) (at the anaerobic threshold (AT)), respiratory compensation point (RCP), and maximum exertion were measured. Random forests (RF) were used to find the variables with the highest importance, which were selected for multiple linear regression (MLR) models. Based on R^2^ and RF variable selection, MLR equations in full, simplified, and the most simplified forms were created for VO_2AT_, HR_AT_, VO_2RCP_, HR_RCP_, VO_2max_, and HR_max_ for CE (R^2^ = 0.46–0.78) and TE (R^2^ = 0.59–0.80). By inputting only HR and power/speed into the RF, MLR models for practical HR calculation on TE and CE (both R^2^ = 0.41–0.75) were created. BC had a significant impact on the majority of CPET parameters. CPET parameters can be accurately predicted between CE and TE testing. Maximal parameters are more predictable than submaximal. Only HR and speed/power from one testing modality could be used to predict HR for another. Created equations, combined with BC analysis, could be used as a method of choice in comprehensive sports diagnostics.

## 1. Introduction

Precise training plans are a key requirement for optimal performance in endurance athletes, allowing us to improve both maximum oxygen uptake (VO_2max_) and competition results [1,2]. Variables obtained from laboratory cardiopulmonary exercise tests (CPET) may also be used to predict race results with considerable accuracy [3]. It is still controversial whether prescription of exercise should rely on heart rate (HR) zones (expressed as %HR_max_) or HR measured at key points, such as the anaerobic threshold (AT) or the respiratory compensation point (RCP), corresponding most closely to critical power [1].

Usually, the modality of testing for CPET is chosen according to the dominant type of exercise performed by the athletes, i.e., treadmill (TE) for runners or cycle ergometer (CE) for cyclists. Results of CPET may vary considerably, depending on the chosen testing modality, likely due to different training experience, muscle activation patterns, and the static component in cycling [4,5,6,7]. A unique challenge is posed by duathletes and triathletes, who train in multiple disciplines, including both cycling and running. Triathlon gained popularity relatively recently, when compared with running or cycling alone; therefore, less research has been conducted on triathletes, often on small groups, and much of the training plan is created based on personal experience of trainers. The traditional ‘more is better’ approach to training, with little training monitoring, is gradually being replaced by sophisticated training monitoring to ensure optimal preparation for events and minimize injury, fatigue, and illness [8]. Training plans are developed with high precision, based on laboratory testing, for elite triathletes, and this trend naturally also affects amateur triathletes. Due to the differences in HR, especially at AT, in cycling and running, it is most likely optimal to use both testing modalities (and, if possible, swimming for triathletes) to prescribe specific training plans. This approach is also used in current research [9]. Testing is also usually carried out several times throughout the season to monitor training and adjust recommendations [10]. A drawback of this approach for the amateur triathletes may be the high cost of CPET testing, as well as the time required to make appointments in a testing center, prepare for each test, and complete it. It would, therefore, be highly practical for triathletes, if it were possible to model running parameters using the results obtained from CE and vice versa.

Assessing the interchangeability of the two testing modalities for the monitoring of triathlons has previously been attempted, and a linear relationship between HR and oxygen consumption (VO_2_) has been observed [10,11]. Other studies also observed a linear relationship between heart rate and VO_2_, but the authors pointed out that large individual variation was observed, which may further reduce the usefulness of this method in sports [12]. However, these relationships do not take into account the different physiology of cycling and running and the large differences in HR, at which the AT occurs, or the differences in lactate levels [13]. Therefore, a simple linear relationship between HR and VO_2_ is likely not sufficient to make the methods interchangeable for modern training prescription, which is often based on HR at various thresholds or critical power, in the case of interval training. This is supported by a more recent study on triathletes, which concluded that separate exercise-specific tests should be used, due to the large differences in HR_AT_ [13].

In this study, we attempt to create regression models to predict exercise parameters in TE (VO_2max_, VO_2AT_, VO_2RCP_, HR_max_, HR_AT_, HR_RCP_, Lac_max_, Lac_AT_, and Lac_RCP_), based on the results of CE, anthropometric data, and analogous models to predict CE parameters.

## 2. Material and Methods

### 2.1. General Study Information and Inclusion Criteria

This was a retrospective study using data recorded from commercial CPET testing in the Sportslab clinic (Warsaw, Poland). This database has also previously been used to study differences between CE and TE in triathletes and to test whether these differences might be explained by anthropometric data [14]. Participants’ results from CPET performed in years 2013–2020 were included in the analysis. Exercise tests were carried out on personal request for optimization of participant’s endurance as a part of prescribed training programs.

The study group consisted of amateur and professional triathletes with a history of taking part in competitive events. Documented intership from the earliest competition start to the day of the first test was an average of 105.1 ± 46.9 months; 95%CI from 96.6 to 113.6. Inclusion criteria were: male gender (due to too few females in the database), age > 18 years, at least 3-month experience in triathlon training, and fulfilling maximum endurance criteria described below. Exclusion criteria were: suffering from any medical condition (both chronic and acute, as well as musculoskeletal disorders or addictions) and taking any medications.

Participants were advised to avoid any exercises at least 24 h prior to the test, eat a light carbohydrate meal, and keep hydrated with isotonic sports drinks 2–3 h before the test. They were instructed to exclude any drugs, caffeine, and cigarettes on the day of testing.

152 participants were finally selected from the database, each with at least one TE and one CE test within 1–52 days of one another. The study group in this paper is an updated and enlarged (an additional 27 males were recruited from January 2021 to November 2021) group from a previous descriptive paper on the differences between CPET parameters in TE and CE [14].

### 2.2. Conditions and Equipment Used during CPET

Every test was preceded by an analysis of body mass (BM) and fat mass (FM), on the body composition (BC) measuring device (Tanita, MC 718, Tokyo, Japan), with the multifrequency 5 kHz/50 kHz/250 kHz electrical bioimpedance method. If the period between both tests was >48 h, the BC measuring has been conducted directly before each of them, and the mean result was used for further analysis. 

BC and CPET took place under the same conditions in the Sportslab Medical Clinic (www.sportslab.pl, Warsaw, Poland, access on 10 December 2021): 40 m^2^ of indoor, air-conditioned space, 40–60% humidity, temperature 20–22 degrees centigrade, and altitude 100 m MSL.

The running test was on a mechanical treadmill (h/p/Cosmos quasar, Nussdorf—Traunstein, Germany), and the cycling test was on a cycle ergonometer Cyclus-2 (RBM elektronik-automation GmbH, Leipzig, Germany). Cardio-pulmonary exertion values were measured by a Cosmed Quark CPET device (Rome, Italy), calibrated before the tests, according to the producer’s instruction. HR indices were taken with the usage of ANT and a chest strap, as a part of the Cosmed Quark CPET equipment (manufacturer’s declared accuracy similar to ECG; ±1 bpm).

### 2.3. Overview of Testing Protocol

Both tests started with a 5 min warm-up, consisting of walking or light pedaling. Initial loads were determined individually to account for participants’ different endurance capacities. The starting power in cycling tests ranged from 60 to 150 W; every 2 min, the load was increased by 20–30 W. The initial treadmill inclination was 1%. A running speed, described individually as a “slow pace”, was selected (ranges between 7 and 12 km/h). Then, the pace increased every 2 min by 1 km/h. 

Athletes were verbally encouraged to maintain the intensity for the longest possible period, in order to assess their maximal aerobic fitness level most accurately. The termination of the test occurred by the operator when the VO_2_ or HR did not increase with higher speed/power or if the participant felt unable to maintain the effort. Cardiopulmonary monitoring was applied during the whole test.

### 2.4. Blood Lactate Examination

A 20 µL blood sample was taken from the fingertip, immediately prior to each CPET, after any change in load or speed, and 3 min after finishing the test. Blood was taken without a break in cycling or running during the CPET. The initial drops of blood were collected into a swab before the proper sample was drawn. A Super GL2 analyzer (Müller Gerätebau GmbH, Freital, Germany) was used to assess lactate concentration (LA). The device was calibrated before each round of analysis.

### 2.5. Final Characteristics of Selected Participants

Results were imported to the Excel file (Microsoft Corporation, Washington, DC, USA). Basic data were anonymized, and analysis was conducted in a custom tool, created in a Python environment, to export AT, RCP, and maximum exertion values from Excel files. According to current standards, CPET results were measured breath-by-breath, with averaged 10 s periods. The highest HR during intervals was recorded, and HR was not averaged [15].

Only cases where at least three of four of the following criteria of maximum exertion were fulfilled were included: (1) RER not smaller than 1.10, (2) achieved VO_2_ plateau (a VO_2_ increase with growth in speed/power lower than 100 mL/min), (3) breathing frequency higher than 45/min, and (4) perceived exertion above 18 in the Borg scale [16].

AT was reached after meeting the following criteria: (1) common start of VE/VO_2_ and VE/VCO_2_ curves, (2) end-tidal partial pressure of oxygen raised constantly with end-tidal partial pressure of carbon dioxide. RCP was reached after meeting the following criteria: (1) PetCO_2_ must decrease after reaching maximal amount, (2) presence of fast nonlinear growth in VE (second deflection), (3) the VE/VCO_2_ ratio achieved minimum and started to rise, and (4) a nonlinear increase in VCO_2_ versus VO_2_ (lack of linearity) [17].

For greater accuracy of calculations, the exact threshold and maximum power/speed were determined based on the linear relationship between the time and rise of power/speed values during the tests. The exact blood lactate concentration, related to speed/power, was estimated for each threshold, based on the lactate measurements and speed/power-time graphs.

### 2.6. Ethical Approval

Due to humans’ involvement, the study was reviewed and obtained approval from the Bioethical Committee-IRB of the Medical University of Warsaw (AKBE/32/2021). Each patient had to provide written informed consent to participate in the study. Procedures have been performed in accordance with the recommendations from the Declaration of Helsinki.

### 2.7. Statistical Analysis

R environment/programming language (version 3.6.4) was applied for statistical analysis. If a lack of results in lactate values was found, imputation was performed with random forests (in seven cases total) [18]. Anderson-Darling test was used to check for normality. Results were calculated as means with standard deviation (SD) and 95% confidence intervals (CI). Maximal power/speed was computed as mean SD (Z-score) for random forest analysis.

To select the variables to include in the MLR models, random forest was used to predict whether the test was performed on TE or CE, based on the CPET results (HR, VO_2_, RER, VE, fR, LA, at the AT, RCP, and maximum). The variables with highest prediction values were selected for further modelling in MLR. It has been demonstrated that RF is one of the most accurate method for variable selection [19,20,21]. Apart from the variables selected with RF, body fat, BMI, and age were also included in the models, as they were previously linked to differences between CPET results in TE and CE [14]. These variables were then used to build predictive models, using multiple linear regression with the brute-force approach (all combinations were tested). In a further attempt to simplify the models (reduce the number of variables), they were also recalculated using only the variables with *p* < 0.05 in the first MLR model. Simple models for practical application were also created by imputing only HR and power/speed into random forest data selection and then creating MLR models with the use of only these variables.

Evaluation of the dependencies in running and cycling scores (dependent variables), body fat (BF), and BMI, with the usage of multiple linear regression (MLR) models. R-squared (R^2^) was used to evaluate the quality of models.

## 3. Results

All the variables displayed normal distribution. The mean differences between groups for the predicted variables are presented in the Appendix A. Participants’ characteristics are shown in Table 1.

Prediction formulae were created based on multiple linear regression results and are presented in Table 2 and Table 3, along with R^2^ and mean absolute error (MAE).

Figure 1 and Figure 2 present graphs of observed vs. predicted results using each of the formulae.

The simple equations for practical use (only HR and speed/power) are presented in Table 4 and Table 5.

The simplified formulae, obtained by removing nonsignificant variables, are presented in the Appendix A.

## 4. Discussion

This is the first study to evaluate and model the transferability of the results between CE and TE in triathletes with the use of multiple linear regression and RF modelling. It is, therefore, impossible to offer a direct comparison of the results with previous literature. The created models display a relatively high R^2^, on average, explaining approximately 70% of the differences between TE and CE.

The simplest formulae, using only HR and power/speed data, also demonstrated fair accuracy, with MAE only slightly lower than the full equations. This offers a possibility to easily estimate HR zones in running after performing a simple HR_max_ test on a cycle ergometer or vice versa, depending on the available equipment. Such equations may be especially useful for amateurs with limited resources and access to equipment.

The obtained MAE are considerably lower for most variables than the actual differences between CE and TE, sometimes by more than 50%, for example, in the case of HR_AT_. Therefore, the formulae offer an alternative to using both tests for triathletes, when performing two tests would be too costly. The estimated values would improve precision of exercise prescription, based on HR_AT_, HR_RCP_, or HR_max_. When considering the precision of the predictive algorithms, it is important to compare it with test–retest reliability of CPET testing. While CPET is the gold standard in testing exercise capacity and considered highly precise, even test results obtained using the same modality, performed by the same team, on the same person, will inevitably vary. Decato et al. recently showed that the mean absolute differences in the CPET results were 7.1% for peak VO_2_ (L·min^−1^), 2.5% for peak HR, 14.7% for VO_2_ at lactate threshold (LT), and 9.2% for HR at LT, with coefficients of variation of 4.9%, 1.8%, 10.4%, and 6.6%, respectively [22]. Previous studies showed similar or higher variance between repeated tests [22,23]. In a meta-analysis, conducted by Vickers et. al., the mean error for VO_2max_ was 2.58 mL·kg^−1^·min^−1^ in the test–retest examination [24]. Considering these results, the MAE observed in this study appear to be acceptable, the prediction error being similar to the test–retest measurement error, where the 5% error is commonly accepted [25]. It is also worth noting that the variance is low for peak parameters but much higher at the aAT and RCP thresholds, possibly due to the difficulty of reliably establishing the exact threshold values. This could also explain the lower R^2^ and higher MAE obtained in our study for submaximal, rather than for maximal parameters.

The lowest prediction accuracy was observed for CE HR_AT_. We speculate that this might be because of the large variation of aerobic capacity in the population, as well as the limited impact of cycling at the AT on the cardiovascular system, relative to the external work performed, which is due to more isolated muscle effort in CE. Therefore, the AT occurs at a lower effort, relative to peak, in effect being less predictable. This hypothesis would require further research to support it. It is also possible that cardiac stroke volume increases, relative to VO_2max_, may be different in CE and TE at different thresholds [26,27,28,29].

RF analysis revealed that the most important variable, in determining whether the results are generated from TE or CE, is VE_AT_. VE_AT_ was also subsequently selected in several of the predictive models, suggesting the importance of different breathing mechanics in CE and TE. It has been shown that CE leads to a greater decrease in respiratory muscle endurance than TE, and it has been suggested that the differences in breathing mechanics may be due to different entrainment of breath in CE and TE [4,30]. Studies showed that triathletes display higher entrainment of breathing in CE than TE and that entrainment decreases with increasing load in CE [31]. The differences in the rates of ventilation may also be due to the size of CO_2_ production, which is dependent on the number of active muscles. Running involves more muscles in the torso and arms, which do not directly contribute to locomotion, as is the case in cycling [26].

Body fat was a significant factor in most of the predictive formulae. It has previously been shown that body fat correlates negatively with aerobic capacity in TE testing (r = −0.89, *p* < 0.001) [32]. It has also been shown that fatness is not directly related to VO_2max_ but has a detrimental effect on submaximal performance in TE [33]. There is little data about the influence of fatness on VO_2max_ in CE, compared to TE. Our results suggest that body fat should have a significantly different effect on CE than on TE, perhaps due to the fact that the additional fat mass increases the load on the treadmill, without contributing significantly to oxygen consumption. Perhaps body fat might also impact breathing differently in TE than in CE, as it has previously been shown that increased body fat is associated with lower lung volumes [34].

## 5. Conclusions

CPET parameters can be accurately (accuracy similar to test–retest error) predicted for a cycle ergometer test, based on treadmill test results and vice versa.

Maximum CPET parameters are more predictable than submaximal parameters.

Even when only HR and speed/power from one testing modality are available, HR may be predicted with fair accuracy for the other testing modality.

Body fat significantly affects the prediction of almost all CPET parameters.

The limitations of the present study were the homogenous ethnicity and gender of the participants (Caucasian males). It was not possible to test swimming, due to a lack of equipment; designing models for swimming could be a future research direction.

## Figures and Tables

**Figure 1 ijerph-19-01830-f001:**
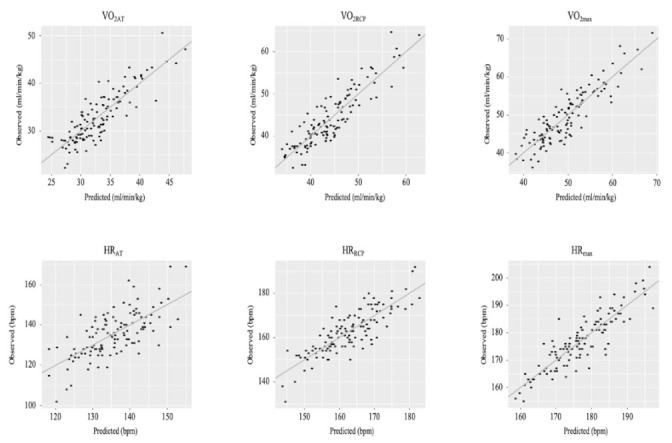
Graphs of observed vs. predicted results using each of the formulae. Abbreviations: VO_2AT_, relative VO_2_ at AT (mL·min^−1^·kg^−1^); VO_2RCP_, relative VO_2_ at RCP (mL·min^−1^·kg^−1^); VO_2max_, relative maximum VO_2_ (mL·min^−1^·kg^−1^); HR_AT_, heart rate at AT (bpm); HR_RCP_, heart rate at RCP (bpm); HR_max_, maximal heart rate (bpm).

**Figure 2 ijerph-19-01830-f002:**
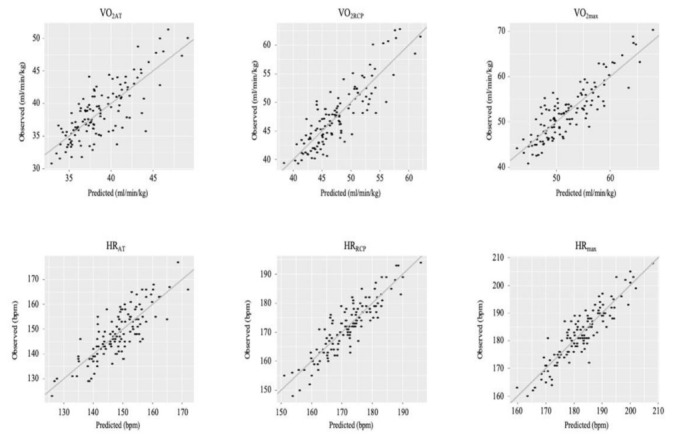
Graphs of observed vs. predicted results using each of the formulae. Abbreviations: VO_2AT_, relative VO_2_ at AT (mL·min^−1^·kg^−1^); VO_2RCP_, relative VO_2_ at RCP (mL·min^−1^·kg^−1^); VO_2max_, relative maximum VO_2_ (mL·min^−1^·kg^−1^); HR_AT_, heart rate at AT (bpm); HR_RCP_, heart rate at RCP (bpm); HR_max_, maximal heart rate (bpm).

**Table 1 ijerph-19-01830-t001:** Participants’ characteristics.

Participants Characteristics				
Variable	Mean	CI		SD
		−95%	95%	
CPET interval (days)	7.28	5.40	9.16	11.74
Age (years)	38.20	36.68	39.73	9.53
Height (cm)	180.64	179.55	181.72	6.77
Weight (kg)	78.31	76.88	79.74	8.92
BMI (kg·m^−2^)	23.97	23.64	24.31	2.10
BF (%)	15.41	14.76	16.05	4.03
FM (kg)	12.28	11.61	12.96	4.19

**Table 2 ijerph-19-01830-t002:** Cycling prediction equations in a full form.

Category	Multiple Regression Equation	MAE	Adjusted R^2^
**VO_2AT_**	20.83_[13.56]SE_ + 0.56_[0.12]SE_x _R_VO_2AT_ − 0.11_[0.05]SE_x _R_VE_AT_ − 0.11_[0.05]SE_x _R_HR_AT_ + 12.06_[9.00]SE_x _R_RER_AT_r + 0.04_[0.03]SE_x _R_fR_RCP_ − 23.13_[12.59]SE_x _R_RER_RCP_ − 0.36_[0.17]SE_x _R_Lac_max_ + 0.30_[0.09]SE_x _R_VO_2max_ + 0.05_[0.05]SE_x _R_HR_max_ + 0.05_[0.03]SE_x _R_VE_RCP_ − 0.13_[0.08]SE_x BF	2.17	0.71
**HR_AT_**	54.84_[30.34]SE_ + 0.41_[0.12]SE_x _R_HR_AT_ − 49.99_[27.99]SE_x _R_RER_RCP_ + 0.41_[0.12]SE_x _R_HR_max_ − 0.30_[0.19]SE_x BF	6.25	0.46
**VO_2RCP_**	−2.39_[5.83]SE_ + 0.62_[0.17]SE_x _R_VO_2AT_ − 0.01_[0.001]SE_x _R_VO_2ATA_ − 0.53_[0.17]SE_x _R_Lac_max_+ 0.32_[0.12]SE_x _R_VO_2max_ + 0.34_[0.18]SE_x _R_VO2_RCP_ + 0.03_[0.03]SE_x _R_VE_RCP_ + 0.52_[0.24]SE_x BMI − 0.32_[0.11]SE_x BF	2.44	0.78
**HR_RCP_**	53.27_[21.96]SE_ + 0.17_[0.11]SE_x _R_HR_AT_ + 0.33_[0.20]SE_x _R_HR_RCP_ − 38.36_[19.97]SE_x _R_RER_RCP_ + 0.40_[0.16]SE_x _R_HR_max_ − 0.42_[0.13]SE_x BF	4.47	0.68
**VO_2max_**	11.75_[4.59]SE_ − 0.05_[0.03]SE_x _R_VE_AT_ + 0.07_[0.03]SE_x _R_fR_RCP_ − 0.35_[0.18]SE_x _R_Lac_max_ + 0.44_[0.12]SE_x _R_VO_2max_ + 0.50_[0.14]SE_x _R_VO_2RCP_ − 0.33_[0.09]SE_x BF	2.55	0.78
**HR_max_**	16.51_[11.24]SE_ + 0.26_[0.13]SE_x _R_HR_RCP_ + 0.57_[0.27]SE_x _R_Lac_max_ + 0.60_[0.12]SE_x _R_HR_max_ + 0.33_[0.31]SE_x BMI − 0.45_[0.15]SE_x BF	3.31	0.78

Abbreviations: CI, confidence interval; SD, standard deviation; CPET, cardiopulmonary exercise test; BMI, body mass index (kg·m^−2^); BF, body fat (%); FM, fat mass (kg).

**Table 3 ijerph-19-01830-t003:** Running prediction equations in a full form.

Category	Multiple Regression Equation	MAE	Adjusted R^2^
**VO_2AT_**	−3.59_[11.10]SE_ − 0.09_[0.05]SE_x _C_VE_AT_ + 0.10_[0.06]SE_x _C_fR_AT_ + 0.002_[0.002]SE_x _C_VO_2ATA_ − 0.21_[0.16]SE_x _C_Lac_max_ + 0.06_[0.03]SE_x _C_HR_max_ + 0.43_[0.05]SE_x _C_VO_2RCP_ + 12.46_[7.90]SE_x _C_RER_max_ − 0.04_[0.03]SE_x Age	2.05	0.59
**HR_AT_**	59.44_[16.41]SE_ − 1.21_[0.40]SE_x _C_VO_2AT_ − 0.24_[0.09]SE_x _C_VE_AT_ + 0.17_[0.13]SE_x _C_HR_AT_ + 0.01_[0.04]SE_x _C_VO_2ATA_ + 0.27_[0.22]SE_x _C_HR_RCP_ + 0.05_[0.03]SE_x _C_P_AT_ − 0.44_[0.28]SE_x _C_VO_2max_ + 0.34_[0.17]SE_x _C_HR_max_ + 0.81_[0.39]SE_x _C_VO_2RCP_ − 1.36_[0.48]SE_x BMI + 0.56_[0.23]SE_x BF	4.66	0.63
**VO_2RCP_**	13.88_[2.60]SE_ + 0.43_[0.14]SE_x _C_VO_2AT_ − 0.22_[0.07]SE_x _C_VE_AT_ + 0.28_[0.11]SE_x _C_fR_AT_ + 0.03_[0.02]SE_x _C_VE_max_ + 0.04_[0.01]SE_x _C_P_AT_ − 0.15_[0.09]SE_x _C_fR_RCP_ + 0.30_[0.10]SE_x _C_VO_2max_ + 0.06_[0.04]SE_x _C_VE_RCP_	2.18	0.73
**HR_RCP_**	12.83_[2.11]SE_ + 0.60_[0.04]SE_x _C_VO_2max_ + 0.03_[0.01]SE_x _C_P_AT_	2.28	0.71
**VO_2max_**	101.31_[26.95]SE_ − 1.12_[0.32]SE_x _C_VO_2AT_ − 0.16_[0.09]SE_x _C_HR_AT_ − 0.07_[0.02]SE_x _C_VE_ma_ + 0.01_[0.002]SE_x _C_VO_2ATA_ + 9.70_[7.73]SE_x _C_RER_AT_ + 0.30_[0.16]SE_x _C_HR_RCP_ − 40.07_[25.14]SE_x _C_RER_RCP_ − 0.39_[0.20]SE_x _C_VO_2max_ + 0.63_[0.13]SE_x _C_HR_max_ + 0.84_[0.28]SE_x _C_VO_2RCP_ − 0.13_[0.05]SE_x Age − 1.11_[0.34]SE_x BMI + 0.32_[0.17]SE_x BF	3.25	0.78
**HR_max_**	28.49_[5.23]SE_ + 0.39_[0.15]SE_x _C_VO_2AT_ − 0.06_[0.03]SE_x _C_HR_AT_ − 0.21_[0.08]SE_x _C_fR_AT_ − 0.003_[0.001]_x _C_VO_2ATA_ − 0.03_[0.01]_x _C_P_AT_ − 0.14_[0.06]_x _C_fR_RCP_ + 0.49_[0.09]SE_x _C_VO_2max_ − 0.07_[0.03]SE_x Age	2.38	0.74
	112.46_[29.03]SE_ − 0.58_[0.26]SE_x _C_VO_2AT_ + 0.07_[0.06]SE_x _C_VE_AT_ − 0.06_[0.02]SE_x _C_VE_max_ + 8.49_[7.51]SE_x _C_RER_AT_ − 57.77_[27.32]SE_x _C_RER_RCP_ + 0.51_[0.26]SE_x _C_Lac_max_ + 0.82_[0.05]SE_x _C_HR_max_ + 0.26_[0.21]SE_x _C_VO_2RCP_ − 0.14_[0.05]SE_x Age − 0.77_[0.32]SE_x BMI + 0.35_[0.16]SE_x BF	3.27	0.80

Abbreviations: MAE, mean absolute error; VO_2AT_, relative VO_2_ at AT (mL·min^−1^·kg^−1^); SE, standard error; _C_VE_AT_, cycling pulmonary ventilation at AT (L·min^−1^); _C_fR_AT_, cycling respiratory rate at AT (breaths per minute); _C_VO_2ATA_, cycling absolute VO_2_ at AT (mL·min^−1^); _C_Lac_max_, cycling maximal lactate concentration (mmol·L^−1^); _C_HR_max_, cycling maximal heart rate (bpm); _C_VO_2RCP_, cycling relative VO_2_ at RCP (mL·min^−1^·kg^−1^); _C_RER_max_, cycling maximal respiratory exchange ratio; age, age (years); HR_AT_, heart rate at AT (bpm); _C_VO_2AT_, cycling relative VO_2_ at AT (mL·min^−1^·kg^−1^); _C_HR_AT_, cycling heart rate at AT (bpm); _C_HR_RCP_, cycling heart rate at RCP (bpm); _C_P_AT_, cycling power at AT (watt); _C_VO_2max_, cycling relative maximum VO_2_ (mL·min^−1^·kg^−1^); BMI, body mass index (kg·m^−2^); BF, body fat (%); VO_2RCP_, relative VO_2_ at RCP (mL·min^−1^·kg^−1^); _C_VE_max_, cycling maximal pulmonary ventilation (L·min^−1^); _C_fR_RCP_, cycling respiratory rate at RCP (breaths per minute); _C_VE_RCP_, cycling pulmonary ventilation at RCP (L·min^−1^); HR_RCP_, heart rate at RCP (bpm); _C_RER_AT_, cycling respiratory exchange ratio at AT; _C_RER_RCP_, cycling respiratory exchange ratio at RCP; VO_2__max_, relative maximum VO_2_ (mL·min^−1^·kg^−1^); HR_max_, maximal heart rate (bpm).

**Table 4 ijerph-19-01830-t004:** Simplified cycling prediction equations.

Category	Multiple Regression Equation	MAE	Adjusted R^2^
**HR_AT_**	20.64_[13.77]SE_ + 0.71_[0.08]SE_x _R_HR_AT_ + 0.83_[0.59]SE_x _R_S_AT_	6.66	0.41
**HR_RCP_**	5.89_[11.24]SE_ + 0.88_[0.06]SE_x _R_HR_RCP_ + 0.52_[0.41]SE_x _R_S_AT_	4.63	0.65
**HR_max_**	16.41_[9.27]SE_ + 0.86_[0.05]SE_x _R_HR_max_ + 0.31_[0.33]SE_x _R_S_AT_	3.66	0.75

Abbreviations: MAE, mean absolute error; HR_AT_, heart rate at AT (bpm); SE, standard error; _R_HR_AT_, running heart rate at AT (bpm); _R_S_AT_, running speed at AT (km·h^−1^); HR_RCP_, heart rate at RCP (bpm); _R_HR_RCP_, running heart rate at RCP (bpm); HR_max_, maximal heart rate (bpm); _R_HR_max_, running maximal heart rate (bpm).

**Table 5 ijerph-19-01830-t005:** Simplified running prediction equations.

Category	Multiple Regression Equation	MAE	Adjusted R^2^
**HR_AT_**	73.98_[9_._05]SE_ + 0.60_[0_._07]SE_x _C_HR_AT_ − 0.04_[0_._02]SE_x _C_P_AT_	5.89	0.41
**HR_RCP_**	54.69_[8_._33]SE_ + 0.76_[0_._05]SE_x _C_HR_RCP_ − 0.03_[0_._02]SE_x _C_P_AT_	4.28	0.66
**HR_max_**	30.84_[8_._58]SE_ + 0.88_[0_._05]SE_x _C_HR_max_ − 0.02_[0_._01]SE_x _C_P_AT_	3.71	0.75

Abbreviations: MAE, mean absolute error; HR_AT_, heart rate at AT (bpm); SE, standard error; _C_HR_AT_, cycling heart rate at AT (bpm); _C_P_AT_, cycling power at AT (watt); HR_RCP_, heart rate at RCP (bpm); _C_HR_RCP_, cycling heart rate at RCP (bpm); HR_max_, maximal heart rate (bpm); _C_HR_max_, cycling maximal heart rate (bpm).

## Data Availability

The raw data supporting the conclusions of this article will be made available by the authors, without undue reservation.

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
