# Peer review of "Transferability of Cardiopulmonary Parameters between Treadmill and Cycle Ergometer Testing in Male Triathletes—Prediction Formulae"

_ijerph, 2022, doi:10.3390/ijerph19031830_

Round 1

Reviewer 1 Report

The authors studied the transferability of results between CE and TE in triathletes with the use of multiple linear regression and RF modeling. The manuscript is interesting and well-written. The study is however limited by the lack of female athletes' inclusion in the study. The authors should have included female athletes too in their cohort of study population no matter how few were the females in the database. I would recommend changing the title of the study and including 'male triathletes' rather than 'athletes' only in the title.

Reviewer 2 Report

A very good study that demonstrate the associations between the tests.

References (specifically the name of journal need to be consistent; i.e. capital or small letters).

Reviewer 3 Report

Is it possible that amateurs and professional treathletes have a different effort behaviour and therefore should be separated, or did you find  comparable results? Please specify.
